# Social and mental health impact of COVID-19 pandemic among health professionals of Gandaki Province, Nepal: A mixed method study

Shishir Paudel[1], Sujan Poudel[2], Dhurba Khatri[3], Anisha Chalise[4]*, Sujan Babu Marahatta[5]

1 Department of Public Health, CiST College, Pokhara University, Kathmandu, Nepal, 2 Nobel College, Pokhara University, Kathmandu, Nepal, 3 School of Health and Allied Sciences, Pokhara University, Lekhnath, Kaski, Nepal, 4 Center for Research on Environment, Health and Population Activities (CREHPA), Lalitpur, Nepal, 5 Manmohan Memorial Institute of Health Sciences, Kathmandu, Nepal

* anisha.chalise90@gmail.com

**Data Availability Statement:** All relevant data are within the manuscript and its Supporting information files.

## Abstract

### Background

The frontline health workers are the key players in the fight against the COVID-19 pandemic, however, several incidences of attacks, stigmatization, and discrimination towards them have been reported throughout the world during the peak of infection. The social impact experienced by health professionals can alter their efficiency and also lead to mental distress. This study aimed to examine the extent of social impact experienced by health professionals currently working in Gandaki Province, Nepal along with the factors associated with their depression status.

### Methods

This was a mixed-method study where a cross-sectional online survey was executed among 418 health professionals followed by in-depth interviews with 14 health professionals of Gandaki Province. The bivariate analysis and multivariate logistic regression were performed to identify the factors associated with depression at 5% level of significance. The information collected from the in-depth interviews was clustered into themes by the researchers.

### Results

Out of 418 health professionals, 304 (72.7%) expressed that COVID-19 has impacted their family relationships, whereas 293 (70.1%) expressed that it impacted their relationships with friends and relatives, and 282 (68.1%) expressed it impacted their relationships with community people. The prevalence of depression among health professionals was noted at 39.0%. Being a female (aOR:1.425,95% CI:1.220–2.410), job dissatisfaction (aOR:1.826, 95% CI:1.105–3.016), COVID-19 impact on family relation (aOR:2.080, 95%

**Funding:** The author(s) received no funding for this work.

**Competing interests:** The authors have declared that no competing interests exist.

CI:1.081–4.002), COVID-19 impact on relationship with friends and relatives (aOR:3.765, 95% CI:1.989–7.177), being badly treated (aOR:2.169, 95% CI:1.303–3.610) and experiencing moderate (aOR:1.655, 95% CI:1.036–2.645) and severe fear (aOR:2.395, 95% CI:1.116–5.137) of COVID-19 were found to the independent predictors of depression. It was noted that the pandemic has an effect on the social relations of health professionals in multiple ways.

## Conclusion

This study noted that there is a significant impact of COVID-19 on health professionals in terms of their social and mental health aspects. The social impact experienced by health professionals is an important predictor of their mental health. The mental health and well-being of these vital workforces can be enhanced by focusing on the social aspect during the pandemic.

## Introduction

The coronavirus disease (COVID-19) which emerged as an outbreak of pneumonia of an unknown etiology in the Hubei Province of China on December 2, 2019 spread rapidly throughout the world and became a public health emergency of international concern within a month [1, 2]. On March 11, 2020, after infecting 118,319 global citizens and claiming 4,292 lives, it was declared a global pandemic [2]. Even after two years of its emergence, though rapid vaccination and strict public contingency plans have helped to reduce the number of cases, the world still struggles to combat COVID-19.

The frontline healthcare providers are the key players in this fight against the COVID-19 crisis as they risk their lives while performing their duties due to a higher risk of exposure to the virus, heavy workload and burnout as well as long working hours [3, 4]. Despite their efforts towards the improvement of health of the general population, several incidences of attacks, stigmatization, and discrimination towards the health workforce have been reported throughout the world [5–9]. Taking this into consideration the World Health Organization had also provided certain guidelines to prevent and address social stigma resulting due to COVID-19 crisis [10]. Moreover, the frontline health care providers in lower-resource settings are denied leave and lack adequate personal protective equipment and remunerations [3, 11, 12]. A meta-analysis assessing the psychological distress among the healthcare providers in Asia noted a pooled prevalence of depression, anxiety, and stress to be 37.5%, 39.7% and 36.4% respectively [13].

In context of Nepal, the first wave of COVID-19 was reported in May 2020, with the first case being identified on January 23, 2020. The cases increased rapidly during the month of June and July and dropped in the month of October 2020 with a total of 62,797 confirmed cases and 401 reported deaths [14–16]. Similarly, the second wave of COVID-19 started in January 2021, peaked in May 2021 and lasted till August 2021. The second wave was most devastating due to the rapid increase in the daily cases and fatalities. In the period of eight months, a total of 772,000 cases and 10,274 deaths were reported [14, 17]. In Gandaki Province alone, a total of 745 cases were reported during first wave, while 2,172 cases were reported during the second wave [18]. The rate of infection and deaths is still actively reported by the Ministry of Health and Population of Nepal [14]. During these first and second phases, the health workers

were reported to have experienced multiple challenges such as shortage of personal protective equipment, essential medicines, lack of training and incentives, high workloads and long working hours leading to physical and mental exhaustion, increased risk of transmission of infection to themselves and their families, as well as stigma and discrimination from communities [12, 19, 20].

It was reported that 1,396 health workers were infected, and 10 health workers died due to the infection during the first wave. During the second wave, 2,974 health workers were infected, and 37 died due to COVID-19 [14]. Several literatures have highlighted the mental health concerns observed during COVID-19 crisis in Nepal among different population groups such as the general public, chronic patients, and mostly healthcare professionals [21–27]. These studies suggest that the rate of depression among healthcare professionals is comparatively higher than that of other population subgroups in Nepal [26–29]. In developing countries like Nepal where there are inadequate health resources, poor service delivery, and community dissatisfaction towards the health system, healthcare providers are often blamed for the outbreaks and they are stigmatized due to their high risk of acquiring COVID-19 infection [30–32].

During the first wave of infection, some Nepalese newspaper articles highlighted the discrimination faced by healthcare workers, including hospital and laboratory staffs in different parts of the country. News reports even stated that healthcare providers faced difficulties finding food, shelter, and social support in local communities solely due to their profession [6, 30]. It has been suggested that the social impact experienced by the health professionals can alter their level of attention and decision-making capacity, which not only affects their efficiency to tackle the outbreaks but might lead to several mental health consequences [32, 33]. While some of the literatures have highlighted the possibility of discrimination among health workers in Nepal, there remains a significant gap in our understanding of the magnitude of social impacts, such as the influence of the pandemic on the health workers' family and social relationships. Little attention has been given to the social aspect of the pandemic, such as how it has influenced the social relationship of this vital workforce with their family, friends and communities. Thus, the aim of this study was to investigate the extent of social impact experienced by health professionals currently working in the Gandaki Province of Nepal, and to examine its relationship with their mental health, using an online survey. Additionally, the study seeks to identify the real experiences of these health professionals and their coping mechanisms through in-depth interviews.

## Methods

### Study design and setting

A mixed-method approach was adopted, where a quantitative survey was followed by a qualitative study. With the aim to assess the extent of social impact experienced by health professionals due to COVID-19, a cross-sectional online survey was executed among 418 health professionals actively working in different fields of health in Gandaki Province. Based on the initial findings from the online survey, the in-depth interview guidelines were developed and a qualitative study was conducted among the health professionals engaged at different health organizations inside the province.

### Data collection and analysis

**Phase 1: Quantitative phase.** To understand the extent of the problem, a quantitative survey was executed through the self-administered questionnaires developed and administrated using Google Forms. As the study was executed during the peak of the outbreak, the link to the

electronic survey form was distributed to health professionals through social media platforms such as Facebook, LinkedIn, Twitter, and Email.

**Sample size and sampling technique.**   The sample size was determined using Cochran's formula for estimation of a proportion ($n = z^2pq/d^2$). A web-based survey conducted during the early phase of the pandemic in Nepal reported 37.5% of the health professionals to have illustrated the symptoms of depression [27]. So, using this past prevalence at 5% allowable error and 95% CI, the initially estimated sample size was 363 health professionals, which was optimized to 400 after adjusting 10% non-response rate. Snowball sampling technique was adopted, where the health professionals were requested to share the Google Form with their contacts working in the Province. All health professionals working in Gandaki Province during the time of data collection were eligible to participate in this study regardless of their level of work, academic qualification, or nature of involvement in COVID-19 response and management. The online data collection was executed during the months of March to August 2021. A total of 418 complete responses were collected from nine districts of Gandaki Province, so the team decided to keep all the complete responses for the analysis. (Table 1).

The online survey questionnaire was divided into five sections, where the first section included questions regarding participants' demographic and academic profile. Second section consisted of questions regarding the change in social relations experienced by the participants at the time of COVID-19. Third section consisted of job satisfaction scale previously validated among the health professionals of Gandaki Province with a Cronbach's alpha of 0.70 [34]. Fourth section consisted of two standard tools, COVID-19 fear scale [35] to assess the level of fear and Patient Health Questionnaire (PHQ-9) [36] to assess the level of depression among health professionals. The PHQ-9 has been translated and validated in Nepali language with a sensitivity of 0.94 and specificity of 0.80 to assess depression at the cut-off of $\geq 10$ [37].

The questions to assess the social impact of COVID-19 pandemic were prepared by the authors in consultation with the experts from public health and social sciences to cover all the crucial aspects of social impact observed in the national context during the pandemic. The questions were converted to an online format using Google Forms and piloted among some health professionals of Bagmati Province. The health professionals provided their input on the length, and relevance of each item to assess the change that they experienced in their social relationships. Following their feedback, an agreement was reached on a final version of the questionnaire (S1 File). The Google Form with the complete set of questions was then pre-tested among another group of healthcare professionals of Bagmati Province and slightly rephrased based on their responses.

**Table 1. Frequency distribution of response collected from each district.**

| Employed District | Number | Proportion |
|---|---|---|
| Baglung District | 56 | 13.4 |
| Gorkha District | 36 | 8.6 |
| Kaski District | 115 | 27.5 |
| Lamjung District | 37 | 8.9 |
| Myagdi District | 22 | 5.3 |
| Nawalpur District | 27 | 6.5 |
| Parbat District | 39 | 9.3 |
| Syangja District | 52 | 12.4 |
| Tanahun District | 34 | 8.1 |
| **Total** | 418 | 100% |

**Quantitative data analysis.** The data acquired through the online survey was exported to Statistical Package for Social Sciences (SPSS) version 22 for analysis (S1 Data). Descriptive statistics were used to describe participants' demographic and academic profile, social experience at the time of COVID-19 pandemic along with their job satisfaction and mental health status in the form of frequencies and percentages. Chi-square test and unadjusted odds ratio was calculated at 5% level of significance to identify the factors associated with depression. The factors found to have statistical significance (p<0.05) in chi-square test were subjected to the final model of multivariable logistic regressions along with the social impact-related variables. Before going into the multivariable analysis, the multi-collinearity among independent variables was tested using Variance Inflation Factor (VIF), where a VIF greater than five was taken as an indication of multi-collinearity between the independent variables. Hosmer and Lemeshow test was used to assess the goodness of fit of the model where p<0.05 represent that model is a poor fit.

**Phase 2: Qualitative phase.** In context of qualitative survey, a total of 14 health professionals from different health organizations actively contributing for the management of COVID-19 crisis within the province were approached for in-depth interviews. The health professionals were selected purposively for the interviews based on their field of expertise, and range of characteristics such as age, ethnicity, gender, and professional roles as well as their acceptance for the face-to-face interviews. (Table 2). The number of interviewees were determined by data saturation; whereby subsequent interviews did not yield any new information.

**Qualitative data collection and analysis.** The consolidated criteria for reporting qualitative studies was followed (S1 Checklist). The interview guideline was developed by two authors (SP and AC) based on the findings from the quantitative phase and expert opinion from the last author (SBM). It was piloted among three healthcare professionals and was amended by SUP based on their feedback.

All fourteen interviews were conducted in-person with the interviewees at their residence by first four authors, following the COVID-19 precautions of distancing, and using sanitizer and mask. All the participants agreed to the face-to-face interview and provided their complete response to all the questions. The interviews were recorded after acquiring informed consent from the participants and field notes were also taken during the interviews. The average duration of the interview was one and half hours. The interview guideline used in the interview is

**Table 2. Characteristics of participants involved in in-depth interview (n = 14).**

| PN | Gender | Position | Experience in Health Sector |
|----|--------|----------|------------------------------|
| 1 | Male | Public Health Inspector | 5 years |
| 2 | Male | Public Health Administrator | 5 years |
| 3 | Female | Medical Officer | 2 years |
| 4 | Male | Founder of a local NGO | 10 years |
| 5 | Male | Radiologist | 3 years |
| 6 | Male | Program Manager | 5 years |
| 7 | Female | Dental Hygienist | 1 year |
| 8 | Male | Health Assistant | 6 years |
| 9 | Female | Nurse | 8 years |
| 10 | Female | Lab Technician | 4 years |
| 11 | Female | Nurse | 6 months |
| 12 | Male | Medical Officer | 3 years |
| 13 | Female | Nurse | 2 years |
| 14 | Female | Pharmacist | 7 years |

attached as a (S2 File). All the interviewers (SP, SUP, DK, AC) are public health postgraduates with past experiences in both quantitative and qualitative data collection and analysis.

All in-depth interview audio were transcribed verbatim on paper by the researchers (SP, SUP, DK, and AC). The researchers cross-checked the transcripts for accuracy and language translation consistency from Nepali to English. Prolonged engagement of authors in every phase of the study and peer debriefing of translations of verbatim were done. The transcripts were not sent to the participants for their review. The interview transcripts and notes were read and analyzed several times to understand the emotions and experiences of participants and to become thoroughly familiar with the content. No software was used for the data analysis. Based on the information provided by the participants, the core concepts expressed by each of the participants were extracted by the authors. The inductive thematic analysis technique was used for which two of the authors (SP and SUP) independently generated the initial codes from the significant statements from the participants and searched for the themes within the transcripts. The concepts were clustered into themes based on their similarities to develop a comprehensive description of the participant's expressions. The next two authors (DK and AC) reviewed the codes and collaboratively defined and named the themes. All authors collaboratively reviewed and finalized the themes.

### Ethical considerations

The ethical approval for this study was obtained from the Institutional Review Committee of Manmohan Memorial Institute of Health Science (Registration no: MMIHS-IRC 574). Written informed consent was obtained from all the participants before they participated in the online survey and the face-to-face interviews. The interview session was recorded after obtaining permission from the participants and all the information of the participants revealing their identity was kept confidential using participant code.

## Results

### Quantitative results

Out of 418 health professionals who submitted their complete response to participate in the quantitative phase, 304 (72.7%) expressed COVID-19 has impacted their family relation positively and/or negatively. Likewise, 293 (70.1%) expressed COVID-19 has impacted their relationship with friends and relatives and 282 (68.1%) expressed COVID-19 has impacted their relationship with community people. Similarly, 107 (25.6%) health professionals reported to have experienced maltreatment by the community people during the pandemic, solely because of their profession. (Table 3).

When subjected to multiple choices to express their perspective on the social impact of COVID-19 that they experienced in their lives, out of all 418 health professionals, 137 reported the feeling of being physically distanced from their family members. Likewise, 91 reported being discriminated by friends and relatives and 131 expressed the feeling of discrimination from community people. (Table 4).

The prevalence of depression among health professionals was noted at 39.0% with mild, moderate, and severe depression at 25.8%, 8.6%, and 4.5% respectively (Table 2). Moreover, 82 (19.6%) health professionals reported to have experienced suicidal ideation in the past two weeks based on PHQ-9 suicidal ideation item. (Table 5).

It was observed that the mean age of the health professionals engaged in the study was 29.49 ±5.52 years, with a minimum and maximum age of 18 and 52 years respectively. Nearly two-fifths (40%) of the participants were nurses followed by 24.9% of health assistants. Almost two-thirds (69.1%) of the health professionals reported to be unsatisfied with their job. The mean

**Table 3. Social impact of COVID-19 experienced by health professionals (n = 418).**

| Variables | Frequency (n) | Percentage (%) |
|---|---|---|
| **Impact of COVID-19 over family relation** | | |
| Yes | 304 | 72.7 |
| No | 114 | 27.3 |
| **Impact of COVID-19 over relationship with friends and relatives** | | |
| Yes | 293 | 70.1 |
| No | 125 | 29.9 |
| **Impact of COVID-19 over relationship community people** | | |
| Yes | 282 | 67.5 |
| No | 136 | 32.5 |
| **Badly Treated by anyone for being a health worker** | | |
| Yes | 107 | 74.4 |
| No | 31 | 25.6 |

score of COVID-19 fear scale was noted at 17.22±5.71 where, moderate and severe fear was noted among 41.6% and 9.1% of the participants respectively, under the cutoff of 17–24 for moderate and ≤25 for severe fear. In bivariate analysis, health professionals' gender, job satisfaction, COVID-19 fear, and social impact of COVID-19 were found to be associated with depression at 5% level of significance. (Table 6).

For multivariate analysis, the Variance Inflation Factor (VIF) test among the independent variables was performed, where the highest reported VIF was 1.841, showing no issue of multi-collinearity. It was observed that the odds of experiencing depression were higher for females (aOR:1.425, 95% CI:1.220–2.410) as compared to their male counterparts. Similarly, the health professionals who were dissatisfied with their job were more at odds of depression

**Table 4. Social impact experienced by the health professional during COVID-19 (n = 418).**

| Variables | Status | |
|---|---|---|
| | Yes (%) | No (%) |
| **Impact of COVID-19 on family relation** | 304 (72.7) | 114 (27.3) |
| *Impacts (Multiple Choices)* (n = 304) | | |
| Feeling distanced from the family members | 137 (45.1) | 167 (54.9) |
| Family seemed more worried about me after COVID-19 | 280 (92.1) | 24 (7.9) |
| Felt emotional closeness with the family members | 221 (72.7) | 83 (27.3) |
| **Impact of COVID-19 over relationship with friends and relatives** | 293 (70.1) | 125 (29.9) |
| *Impacts (Multiple Choices)* (n = 293) | | |
| Friends and relatives expressed fear being nearby | 210 (71.7) | 83 (19.9) |
| Friends and relatives came closer after COVID-19 | 158 (53.9) | 135 (46.1) |
| Discriminated by friends and relatives after COVID-19 | 91 (31.1) | 202 (68.9) |
| Family and relatives were distanced after COVID-19 | 218 (74.4) | 75 (25.6) |
| **Impact of COVID-19 over relationship community people** | 282 (67.5) | 136 (32.5) |
| *Impacts (Multiple Choices)* (n = 282) | | |
| Community people expressed feared being nearby | 190 (67.4) | 92 (32.6) |
| Community people become close after COVID-19 | 89 (31.6) | 193 (68.4) |
| Discrimination from community after COVID-19 | 131 (46.5) | 151 (53.5) |
| Community people distanced after COVID-19 | 210 (74.5) | 72 (25.5) |
| **Badly Treated by anyone for being a health worker** | 107 (74.4) | 31 (25.6) |

**Table 5. Mental distress experienced by the health professionals.**

| Variables | Frequency (n) | Percentage (%) | 95% CI |
|---|---|---|---|
| **Depression Status** | | | |
| Presence | 163 | 39.0 | 34.2–43.9 |
| Absence | 255 | 61.0 | 56.1–65.8 |
| **Depression Level** | | | |
| No Depression | 255 | 61.0 | 56.1–65.8 |
| Mild Depression | 108 | 25.8 | 22.2–30.1 |
| Moderate Depression | 36 | 8.6 | 6.1–11.5 |
| Moderately Severe Depression | 11 | 2.6 | 1.2–4.5 |
| Severe Depression | 8 | 1.9 | 0.7–3.3 |

than those who were satisfied (aOR: 1.826, 95% CI: 1.105–3.016). Health professionals experiencing COVID-19 impact on family relation were twice (aOR: 2.080, 95% CI: 1.081–4.002) more likely to be depressed, while those who experienced the impact of COVID-19 on their relationship with friends and relatives were found to be thrice (aOR: 3.765, 95% CI: 1.989–7.177) more likely to be depressed. Similarly, there was a two-fold (aOR: 2.169, 95% CI: 1.303–3.610) increase in the odds of depression among health professionals who were badly treated by others due to their profession during COVID-19. Fear of COVID-19 was found to be statistically significant with depression as the odds of depression increased among those experiencing moderate (aOR: 1.655, 95% CI: 1.036–2.645) and severe fear (aOR: 2.395, 95% CI: 1.116–5.137) of COVID-19. (Table 7).

## Qualitative results from interviews

**Social impact experienced by health professionals due to COVID-19.** *Theme 1: Change in social relationship.* During our quantitative study, it was noted that more than half of the participants reported experiencing changes in their social relationships with family, friends, and community members during the pandemic. This observation was consistent with our qualitative findings, as many participants reported experiencing such changes in both positive and negative ways. Some of the health professionals expressed that they had been close to their community, providing health information and guidance at the time of infection, while others expressed being forced to be distanced from their community, relatives, and friends. Some of the participants even reported to have voluntarily distanced themselves from their family due to the fear of infecting their loved ones.

*"Many relatives and friends contact me frequently for advice on home isolation, test procedure, days for quarantine and other COVID related information"*

*–IDI-1*

*"I am often contacted by my friends and neighbors to ask about COVID-19 test and things to do if they feel unwell even in case of negative RT-PCR reports."*

*–IDI-4*

*"I live in a rented home near my office with limited space. Sometimes when I go home, I fear I might transfer infection to my family members. This makes me feel if I don't have to go home or if I had bigger space where I could isolate myself."*

*–IDI 2*

**Table 6. Factors associated with depression among health professionals.**

| Variables | n (%) | Depression | | X$^2$ (p-value) |
|---|---|---|---|---|
| | | **Yes** | **No** | |
| **Age** | | | | |
| <20 years | 27 (6.5) | 11 (40.7%) | 16 (59.3%) | 1.525 (0.676) |
| 20–30 years | 225 (53.8) | 92 (40.9%) | 133 (59.1%) | |
| 30–40 years | 148 (35.4) | 52 (35.1%) | 96 (64.9%) | |
| >40 years | 18 (4.3) | 8 (44.4%) | 10 (55.6%) | |
| **Gender** | | | | |
| Female | 237 (56.7) | 105 (44.3%) | 132 (55.7%) | 6.484 (0.011*) |
| Male | 181 (43.3) | 58 (32.0%) | 123 (68.0%) | |
| **Marital Status** | | | | |
| Unmarried | 181 (43.3) | 60 (33.1%) | 121 (66.9%) | 5.182 (0.075) |
| Married | 226 (54.1) | 97 (42.9%) | 129 (57.1%) | |
| Divorced/Widow | 11 (2.6) | 6 (54.5%) | 5 (45.5%) | |
| **Housing** | | | | |
| Rent | 135 (32.3) | 57 (42.2%) | 78 (57.8%) | 1.363 (0.506) |
| Own House | 248 (59.3) | 91 (36.7%) | 157 (63.3%) | |
| Quarters | 35 (8.4) | 15 (42.9%) | 20 (57.1%) | |
| **Education** | | | | |
| TSLC | 27 (6.5) | 8 (29.6%) | 19 (70.4%) | 8.564 (0.056) |
| Diploma | 137 (32.8) | 66 (48.2%) | 71 (51.8%) | |
| Bachelor | 199 (47.6) | 73 (36.7%) | 126 (63.3%) | |
| Masters and above | 55 (13.2) | 16 (29.1%) | 39 (70.9%) | |
| **Organization** | | | | |
| Public organization | 156 (37.3) | 56 (35.9%) | 100 (64.1%) | 4.316 (0.116) |
| Private organization | 240 (57.4) | 102 (42.5%) | 138 (57.5%) | |
| Healthcare business [a] | 22 (5.3) | 5 (22.7%) | 17 (77.3%) | |
| **Job Title** | | | | |
| Nurse | 149 (35.6) | 72 (48.3%) | 77 (51.7%) | 10.591 (0.060) |
| Doctor | 47 (11.2) | 16 (34.0%) | 31 (66.0%) | |
| Health Assistant/ Lab technician | 104 (24.9) | 30 (28.8%) | 74 (71.2%) | |
| Public Health Officer | 14 (3.3) | 5 (35.7%) | 9 (64.3%) | |
| Project Coordinator [b] | 46 (11.0) | 17 (37.0%) | 29 (63.0%) | |
| Pharmacist | 58 (13.9) | 23 (39.7%) | 35 (60.3%) | |
| **Job satisfaction** | | | | |
| Satisfied | 129 (30.9) | 128 (44.3%) | 161 (55.7%) | 11.039 (0.001*) |
| Not Satisfied | 289 (69.1) | 35 (27.1%) | 94 (72.9%) | |
| **Impact over family relation** | | | | |
| Yes | 304 (72.7) | 142 (46.7%) | 162 (53.3%) | 27.892 (<0.001**) |
| No | 114 (27.3) | 21 (18.4%) | 93 (81.6%) | |
| **Impact over relationship with friends and relatives** | | | | |
| Yes | 293 (70.1) | 142 (48.5%) | 151 (51.5%) | 36.929 (<0.001**) |
| No | 125 (29.9) | 21 (16.8%) | 104 (83.2%) | |
| **Impact over relationship community people** | | | | |
| Yes | 282 (67.5) | 127 (45.0%) | 155 (55.0%) | 13.293 (<0.001**) |
| No | 136 (32.5) | 36 (26.5%) | 100 (73.5%) | |
| **Badly Treated for being a health worker** | | | | |

*(Continued)*

**Table 6.** (Continued)

| Variables | n (%) | Depression | | X² (p-value) |
| --- | --- | --- | --- | --- |
| | | Yes | No | |
| Yes | 107 (25.6) | 61 (57.0%) | 46 (43.0%) | 19.618 (<0.001**) |
| No | 311 (74.4) | 102 (32.8%) | 209 (67.2%) | |
| **COVID-19 Fear** | | | | |
| No Fear | 206 (49.3) | 66 (32.0%) | 140 (68.0%) | 9.183 (<0.010*) |
| Moderate Fear | 174 (41.6) | 77 (44.3%) | 97 (55.7%) | |
| Severe Fear | 38 (9.1) | 20 (52.6%) | 18 (47.4%) | |

*statistically significant at p<0.05,

**statistically significant at p<0.001;

[a] Healthcare business includes pharmacy, dental clinic, and clinical stores;

[b] Project coordinators includes: public health professionals engaged in different projects in I/NGOs

The participants also reported to have experienced different forms of discrimination from the family as well as community people. During the interviews, some of the health professionals shared how they were maltreated by their landlords, community people and even local leaders during the time of COVID-19.

**Table 7. Predictors of depression among health professionals.**

| Variables | uOR | 95% CI | aOR | 95% CI |
| --- | --- | --- | --- | --- |
| **Gender** | | | | |
| Female | 1.687 | 1.126–2.526 | 1.425* | 1.220–2.410 |
| Male | Ref | | Ref | |
| **Job satisfaction** | | | | |
| Satisfied | Ref | | Ref | |
| Not Satisfied | 2.135 | 1.358–3.356 | 1.826* | 1.105–3.016 |
| **Impact over family relation** | | | | |
| Yes | 3.882 | 2.298–6.558 | 2.080* | 1.081–4.002 |
| No | Ref | | Ref | |
| **Impact over relationship with friends and relatives** | | | | |
| Yes | 4.657 | 2.764–7.848 | 3.765* | 1.989–7.127 |
| No | Ref | | Ref | |
| **Impact over relationship community people** | | | | |
| Yes | 2.276 | 1.455–3.560 | 1.067 | 0.792–1.445 |
| No | Ref | | Ref | |
| **Badly Treated for being a health worker** | | | | |
| Yes | 2.171 | 1.732–4.262 | 2.169* | 1.303–3.610 |
| No | Ref | | Ref | |
| **COVID-19 Fear** | | | | |
| No Fear | Ref | | | |
| Moderate Fear | 1.684 | 1.108–2.559 | 1.655* | 1.036–2.645 |
| Severe Fear | 2.354 | 1.169–4.750 | 2.395* | 1.116–5.137 |

*Statistical significance at p<0.05; Hosmer Lemeshow Chi-square 11.027, p = 0.200; CI: confidence interval, uOR: Unadjusted odds ratio, aOR: Adjusted odds ratio

*"My house owner said to me that I should either close my clinic or shift from his house because they feel insecure when I am there."*

*–IDI 9*

*"Sometimes I feel like the shopkeepers behave differently around me. They seem to be more conscious when I go to their store to buy something. They always keep their eyes on me and keep a distance and use disinfectant more often than they do around other customers."*

*–IDI 11*

*"I have been asked by my father-in-law to quit my job or take some time off until the rate of infection declines and everything come to normal."*

*- IDI-5*

*Theme 2*: *Psychological distress during the pandemic.* In the quantitative study, it was noted that a significant proportion of healthcare professionals screened positive for depression. Similarly, during the in-depth interviews, it was expressed by most of the professionals that the feeling of social isolation disturbed them emotionally and psychologically. They were disappointed for being assaulted and dishonored by the people while risking their life as frontline workers.

*"The community people blame us if any life is lost. We don't have enough equipment and resources but we are trying. Sometimes all of this lowers my self-esteem; Even though we feel demoralized, we continue to serve as it is the right thing to do."*

*–IDI 4*

*"Due to my exposure, my family was also infected. We went through a lot of hardship, but what did we receive from it? We are the ones who get blamed."*

*–IDI 13*

The health professionals were frustrated by inadequate resources at the time of crisis and pressure from the local leaders to direct scarce resources towards the elites of the community.

*"During the time of vaccination, many of us were pressurized to hide vaccines for families of the local leaders and their people. We knew it wasn't ethical but we had no choice other than to comply with their terms."*

*–IDI 6*

*"We are risking our life to save people and continue to provide services without any rest. But still, we don't have any support system. The leaders don't care about us, nor about the health system. We haven't even received our risk allowance. "*

*–IDI 11*

*Theme 3*: *Coping strategy adopted to handle social discrimination and maltreatment.* During the quantitative survey, it was noted that a significant proportion of health professionals experienced changes in their social relationships. The quantitative findings also suggested that this social impact of the pandemic has a significant association with the mental health of health professionals. The coping mechanisms of the health professionals towards both the change in social relationships and the depressive symptoms resulting from the pandemic were explored

during the qualitative phase of the study. While participants shared about their coping mechanisms to deal with changes in social relationships, there was limited information obtained on their coping strategies to manage mental distress. The qualitative analysis revealed that a common coping strategy among most professionals was ignorance, whereby they tried to overlook the maltreatment they experienced. Some of the health professionals also expressed that, sharing their awful experiences and distress with their loved ones helped them cope with such dissatisfaction, discrimination, and maltreatment.

*"Yes, we are not treated right but what else can we do other than to serve? We choose this profession and it's our responsibility regardless of any unpleasing events."*

–IDI 2

*"My family is really being my support system in these stressful days. COVID has made our interaction more frequent and sharing my day with them calms me and helps me to digest all the distressing events."*

–IDI 9

*"At the beginning, I wasn't worried about the infection but later as the rate of infection increased exponentially and mortality raised each day, I was terrified. It was stressful but as a health worker we have to tackle this crisis so we shouldn't care much about what people say or do, we should just aim to serve our best."*

–IDI 7

*Theme 4*: *Expectations from government and leaders*. During the interviews, considering COVID-19 as a national health emergency, the healthcare professionals were asked about the support they received from all three tiers of the government. They were also asked to provide recommendations to local and provincial governments to support the social and mental well-being of health professionals. The participants expressed dissatisfaction over the response of all three levels of government in the management of COVID-19 crisis, citing issues such as the lack of incentives for frontline workers, insufficient personal protective equipment, absence of insurance coverage, political pressure, and corruption.

*"Incentives and other facilities are for the higher-level staffs who seat in the office even if they don't work at the frontline but the main frontline workforce like us are still not getting any benefits. Neither are we getting any respect from the communities for our efforts."*

–IDI 8

Even in the case of privately owned health organizations, the health professionals were dissatisfied and demotivated by the overall COVID-19 management and response approach. During the interview, the participants notified that some of the organizations deducted the salary of their staff as they were bound to work from home whereas some of the organizations provided their staffs with the services of COVID-19 insurance which covered their basic remuneration. Staffs from some organizations also praised their management for the support during the pandemic. on the contrary, the staffs from some of the private hospitals notified that they did not receive any kind of training and in some cases proper PPE to protect themselves from infection.

*"Our organization is very supportive during this time of pandemic. I feel very happy that the organization had provided the full salary along with some incentive, make the provision of*

*work from home in the meantime, and also provide the different level of training to tackle the pandemic"*

*–IDI 3*

The participants from both government and private organizations had a similar verdict that the government policies are well documented and could improve the level of satisfaction among health providers only if the policies are implemented properly. Proper and smooth implementation of the policies and acts, provision of sufficient medical supplies for diagnosis, and proper incentives for the health workforce were the key recommendations from almost all the participants for the government to tackle the COVID-19 crisis.

*"Policy implementation must be enhanced to improve the healthcare system as many good policies and acts have been formulated but, in action, we sadly have to say there is almost zero implementation and commitment over those well-written documents."*

*–IDI 14*

The role of local government was also highlighted by some health professionals. It was stressed that the local government should focus on the health sector plan and program critically and should consider health as a priority program rather than just focusing on tangible developmental projects. The healthcare professionals also suggested that the Ministry of Health and Population at the Federal and Provincial levels should identify the grassroots problems of health professionals while implementing different programs by consulting with the local government.

*"It is local government's responsibility to regulate or monitor the private hospital and clinics in terms of COVID-19 testing and diagnosis for the effective regulation"*

*–IDI 4*

## Discussion

This study examined the social and mental health impact of COVID-19 among the health professionals working in the Gandaki Province of Nepal at the time of the second wave of the pandemic. It was noted that the pandemic had a significant impact on social relations of the health professionals as almost three-fourths of them had perceived that COVID-19 had influenced their relationships with their family, friends, and community people. Similar to our observation, a study from Saudi Arabia also noted a significant impact of COVID-19 on the social aspect of health professionals' life such as relationship with families and communities, and stigmatization [9]. Likewise, a study among Austrian nurses revealed that they were socially isolated and distanced from their families, which lead to an increase in the use of social media for communication [4].

The significant impact of COVID-19 has also been observed in the mental health of this vital workforce as 39.0% of the health professionals reported to have experienced depressive symptoms. The observed prevalence of depression in our study is near to the depression rate observed in another web-based study from Nepal, where out of 475 healthcare workers, 178 (37.5%) were depressed [27]. A similar higher rate of depression was also observed by another web-based study where 41.3% of the healthcare workers were depressed when subjected to PHQ-9 [26]. The psychosocial and mental health vulnerability of health professionals has been

highlighted throughout the world regardless of the national economy [4, 9, 38]. The rising concern of the mental and social impact of pandemic among the Nepalese population was predicted even at the initial phase of COVID-19 infection in Nepal during the time of first countrywide lockdown [16, 19, 23]. These rates of depression among healthcare professionals are significantly higher than that of the general population of Nepal (8%–35%) [25, 39].

The social and mental health impact among these vital human resources can influence the efficiency and performance of the health system at the time of the pandemic [32, 40, 41]. In-depth interviews revealed that the health professionals were disappointed and their self-esteem had been lowered at the time of pandemic. This has significantly impacted their mental health as well as job satisfaction. Thus, this issue requires specific attention from concerned authorities.

It was observed that female health professionals were at higher odds of experiencing depressive symptoms at this time of crisis. A meta-analysis based on the global prevalence of depression among health workers at the time of COVID-19 pandemic suggested that a gender difference exists in terms of depression. This review revealed that the rate of depression was 32% (95% CI 23%-44%) for females and 23% (95% CI: 16%-32%) for males [42]. In Nepalese society, the social status of women in general is still considered to be inferior to men. During the interviews, the female health professionals reported pressure from their families to quit their job for the safety of the elderly and children which wasn't the case for male professionals.

It was observed that the social impact experienced by the health professionals is an important predictor of depression among these vital human resources as the professionals who experienced change in relation with their family, friends, and community people had higher odds of experiencing depressive symptoms. Past literatures suggest that the perceived social support, discrimination, and mental health outcomes among health workers have been found to be strongly associated with each other at the time of the pandemic [43, 44].

The fear of COVID-19 was also found to be associated with depression symptoms among health professionals while it also impacted the social response from the community people. It was also noted that the fear of COVID-19 infection had significantly impacted the social relation of the health professionals leading them to voluntarily distance themselves from their family and friends. A meta-analysis based on 33 studies suggested that a strong relationship exists between the perceived fear of COVID-19 and the experience of different mental health issues including depression and insomnia [45].

Job satisfaction of the health worker is linked with the quality of care and service provision [46]. In this study, it was also observed that nearly three-fifth of the health professionals were dissatisfied with their job at the time of the pandemic and a statistically significant relationship existed between job satisfaction and depressive symptoms among the professionals. Similar findings of lowered job satisfaction and its relation with mental distress among healthcare professionals at the time of COVID-19 pandemic were observed in different countries [11, 47, 48]. From the interview, it was revealed that most of the health professionals were dissatisfied with their job mostly due to the feeling of being maltreated by the community people and neglected by the health system even while risking their own and their families' lives.

## Limitation

Though this is one of the few studies to assess the social perspective of COVID-19 among health professionals, some of the limitations of this study need to be acknowledged. Firstly, this study was conducted during the second phase of the pandemic, thus the social and mental health impact observed might be influenced by the pre-pandemic state. Secondly, due to the high rate of COVID-19 infection, the quantitative data was collected through the internet,

which might have introduced some selection bias, as the openly shared link might not have been accessed by all the eligible health professionals. In the online survey, some of the health professionals were screened to have moderate to severe depression, but due to the nature of remote data collection, we failed to assess the information about their help-seeking behavior for its treatment. During interviews, the participants were more expressive regarding their coping strategies adopted for their social experiences but gave little information about the strategies adopted to manage their psychological distress.

## Conclusion

The study aimed to examine the social and mental health impact of the COVID-19 pandemic on healthcare professionals of Gandaki Province, Nepal. Both quantitative and qualitative data indicated a significant impact of COVID-19 among health professionals in terms of their social and mental health aspects. Addressing the social impact of the pandemic is crucial for the mental health and well-being of healthcare professionals. Effective coordination across all levels of the health system, including the private sector must be ensured for appropriate management of current and future pandemics.

## Supporting information

**S1 File. Survey questionnaire.**
(DOCX)

**S2 File. Interview guideline.**
(DOCX)

**S1 Data.**
(SAV)

**S1 Checklist. COREQ checklist.**
(DOC)

## Acknowledgments

We share our gratitude to all the health professionals who participated in this study and provided their valuable time and information. Without them, this study wouldn't have been possible.

## Author Contributions

**Conceptualization:** Shishir Paudel, Sujan Poudel, Dhurba Khatri, Anisha Chalise.

**Data curation:** Shishir Paudel, Sujan Poudel, Dhurba Khatri, Anisha Chalise.

**Formal analysis:** Shishir Paudel, Sujan Poudel, Anisha Chalise.

**Investigation:** Shishir Paudel, Sujan Poudel.

**Methodology:** Shishir Paudel, Sujan Poudel, Dhurba Khatri, Anisha Chalise, Sujan Babu Marahatta.

**Supervision:** Sujan Babu Marahatta.

**Validation:** Shishir Paudel, Sujan Poudel, Dhurba Khatri, Anisha Chalise.

**Writing – original draft:** Shishir Paudel, Anisha Chalise.

**Writing – review & editing:** Shishir Paudel, Sujan Poudel, Dhurba Khatri, Anisha Chalise.

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
