## [Decision Letter · Decision Letter 0]

26 Dec 2022

PONE-D-22-30030Social and Mental health impact of COVID-19 Pandemic among Health Professionals of Gandaki Province, Nepal: A mixed method studyPLOS ONE

Dear Dr. Chalise,

Thank you for submitting your manuscript to PLOS ONE. After careful consideration, we feel that it has merit but does not fully meet PLOS ONE’s publication criteria as it currently stands. Therefore, we invite you to submit a revised version of the manuscript that addresses the points raised during the review process.

It’s a good piece of work exploring the social and mental health aspects of health workers of Nepal. While considering the reviewers comments I suggest you rework on the issues raised by reviewers on the following subheadings.

The paper stated about the significant impact in different terms. Is there any test of significance you performed or something else?

Information on regional, global and national context of COVID 19 and mental health been missing. I suggest adding this information in the paper. Little information been provided why health workers are the important players for the COVID 19 and related mental health issues. Similarly, little is stated about types of health workers, are they currently working or those they managed the COVID 19 cases? Furthermore, difference between 1^st^ and 2^nd^ wave of COVID cases in Nepal and their impact is another aspect to deal with. Likewise, it is important to tell what does it mean social and mental health aspect?

Moreover, significant information about Nepal’s COVID 19 response is missing. The author did not consider about the difference between 1^st^ and 2^nd^/3^rd^ wave of COVID 19. Only relied on the information published in newspaper however there are several papers exploring the issues of social and mental health issues in Nepal which is important to review while working on this paper. The authors did not provide any information about of health worker’s death and suffering data in the paper which can be the important for social and mental health of health workers. Therefore, I suggest reviewing such papers which been published form similar setting including Nepal.

While moving to the methodology section, I felt few improvements can be done. The authors did not talk about peak of epidemics in Gandaki province (period, cases and their comparison with national data).  The reasons behind choosing 418 sample size been missing. Similarly, for* Hosmer and Lemeshow* test what is the cut off value you consider the most fit model? How been the interview conducted (Physical or online)?

While reading the paper, I found some typos been occurred. For an example- Consistency form or from Nepali to English. Similarly, details of data analysis technique been missing.

In the result section, consistency in writing the table headings were reported. Therefore, suggested to review similar paper while reporting the table headings and subheadings. Similarly, I suggest you to review the style of reporting of aOR(CI), uOR( CI)- Please refer to other paper.

You have explored the theme 1, 2, 3 etc. but there is little marriage between quantitative and qualitative information. Therefore, suggested having marriage between qualitative and quantities information. So, I suggest reviewing the similar paper on the mixed method studies.

Authors mentioned about the MoHP expectations were explored but little information were provided on provincial MoHP or federal MoHP. At the point the authors can provide information about how the health system worked or suffered in COVID 19 response in federal context.

Similar improvements on discussion and conclusion section are required. Please try to incorporate the reviewer’s comments. I further suggest providing all the supplementary materials such as questionnaire, checklist, consent form, reporting guideline etc. These supplementary files can be used as in example.

Example: - data was collected through the standard checklist (Supplementary file A).

We look forward to receiving your revised manuscript.

Kind regards,

Kanchan Thapa, MPH, MPhil

Academic Editor

PLOS ONE

Journal Requirements:

Additional Editor Comments:

Dear Authors,

It’s a good piece of work exploring the social and mental health aspects of health workers of Nepal. While considering the reviewers comments I suggest you rework on the issues raised by reviewers on the following subheadings.

The paper stated about the significant impact in different terms. Is there any test of significance you performed or something else?

Information on regional, global and national context of COVID 19 and mental health been missing. I suggest adding this information in the paper. Little information been provided why health workers are the important players for the COVID 19 and related mental health issues. Similarly, little is stated about types of health workers, are they currently working or those they managed the COVID 19 cases? Furthermore, difference between 1st and 2nd wave of COVID cases in Nepal and their impact is another aspect to deal with. Likewise, it is important to tell what does it mean social and mental health aspect?

Moreover, significant information about Nepal’s COVID 19 response is missing. The author did not consider about the difference between 1st and 2nd/3rd wave of COVID 19. Only relied on the information published in newspaper however there are several papers exploring the issues of social and mental health issues in Nepal which is important to review while working on this paper. The authors did not provide any information about of health worker’s death and suffering data in the paper which can be the important for social and mental health of health workers. Therefore, I suggest reviewing such papers which been published form similar setting including Nepal.

While moving to the methodology section, I felt few improvements can be done. The authors did not talk about peak of epidemics in Gandaki province (period, cases and their comparison with national data). The reasons behind choosing 418 sample size been missing. Similarly, for Hosmer and Lemeshow test what is the cut off value you consider the most fit model? How been the interview conducted (Physical or online)?

While reading the paper, I found some typos been occurred. For an example- Consistency form or from Nepali to English. Similarly, details of data analysis technique been missing.

In the result section, consistency in writing the table headings were reported. Therefore, suggested to review similar paper while reporting the table headings and subheadings. Similarly, I suggest you to review the style of reporting of aOR(CI), uOR( CI)- Please refer to other paper.

You have explored the theme 1, 2, 3 etc. but there is little marriage between quantitative and qualitative information. Therefore, suggested having marriage between qualitative and quantities information. So, I suggest reviewing the similar paper on the mixed method studies.

Authors mentioned about the MoHP expectations were explored but little information were provided on provincial MoHP or federal MoHP. At the point the authors can provide information about how the health system worked or suffered in COVID 19 response in federal context.

Similar improvements on discussion and conclusion section are required. Please try to incorporate the reviewer’s comments. I further suggest providing all the supplementary materials such as questionnaire, checklist, consent form, reporting guideline etc. These supplementary files can be used as in example.

Example: - data was collected through the standard checklist (Supplementary file A).

Reviewers' comments:

Reviewer's Responses to Questions

**Comments to the Author**

1. Is the manuscript technically sound, and do the data support the conclusions?

Reviewer #1: Yes

Reviewer #2: Yes

2. Has the statistical analysis been performed appropriately and rigorously? 

Reviewer #1: Yes

Reviewer #2: Yes

3. Have the authors made all data underlying the findings in their manuscript fully available?

Reviewer #1: Yes

Reviewer #2: Yes

4. Is the manuscript presented in an intelligible fashion and written in standard English?

Reviewer #1: Yes

Reviewer #2: Yes

5. Review Comments to the Author

Reviewer #1: The manuscript with the number PONE-D-22-30030 and the title "Social and Mental health impact of COVID-19 pandemic among Health Professionals of Gandaki Province, Nepal: A mixed method study" reports on correlations of depression with various predicting factors. The study is well written and has interesting results due to the mixed-methods design. Language and structure of the paper are of very good quality and the paper reads well and smoothly. Statistical testing and interpretation of the results are appropriate. While the questionnaire includes many interesting aspects of social influences on depression rate, I miss any question about seeking help for the depression. Especially, when some of the health workers report severe depression, it would be of high interest what kind of treatment they are seeking. Especially for the results and the discussion the treatment and adherence to treatment in Nepal would be highly interesting and should be included. Additionally, depression rates in the general population in Nepal are necessary to compare to the health workers. This could end in an odds ratio or risk ratio calculation and would be an highly informative addition.

Detailed comments:

The citation style has to be adapted to PLOS One style.

In Table 6 delete the "0" before Impact over family relation.

Reviewer #2: If you used valid questionnaire, please mention validity and reliability of them , if not please mention it in limitations.

Please clarify: What method was used for the qualitative study? What was the way of coding and reaching the themes? What methods are used for the validity of the qualitative study?

Abbreviations should be given in full words the first time they are used.

correct some writing errors e.g., font size in the paragraph after table 5 and removing 0 from the word impact in the table 6

6. PLOS authors have the option to publish the peer review history of their article (what does this mean?). If published, this will include your full peer review and any attached files.

Reviewer #1: **Yes: **Markus Boeckle

Reviewer #2: No

---

## [Author Response · Author response to Decision Letter 0]

27 Feb 2023

Editor Comments and Reply: 

The paper stated about the significant impact in different terms. Is there any test of significance you performed or something else?

For the quantitative data, Chi-square test and Binary logistic regression has been performed as the test of significance at 5% level of significance. 

Information on regional, global and national context of COVID 19 and mental health been missing. I suggest adding this information in the paper. 

Thank you for your suggestion. We have added some information in these aspects in the Introduction section. 

Little information been provided why health workers are the important players for the COVID 19 and related mental health issues. 

Thank you for your suggestion. We have added some information addressing this in the second paragraph of Introduction section. 

Similarly, little is stated about types of health workers, are they currently working or those they managed the COVID 19 cases? 

The health professionals involved in this study were all those who were actively working for the overall healthcare system, and not just those engaged in managing the COVID cases. We have tried to add some statements in the methods section under sample size and sampling technique to clarify this issue. 

Furthermore, difference between 1st and 2nd wave of COVID cases in Nepal and their impact is another aspect to deal with. 

We have added about the context of 1st and 2nd wave of COVID cases in Nepal in 3rd and 4th paragraphs of Introduction section. 

Likewise, it is important to tell what does it mean social and mental health aspect?

Under the Introduction section, we have tried to clarify what we meant by social aspect of the pandemic. We have also attached our questionnaire as a supplementary file to better clarify this issue.

Moreover, significant information about Nepal’s COVID 19 response is missing. The author did not consider about the difference between 1st and 2nd/3rd wave of COVID 19. Only relied on the information published in newspaper however there are several papers exploring the issues of social and mental health issues in Nepal which is important to review while working on this paper. The authors did not provide any information about of health worker’s death and suffering data in the paper which can be the important for social and mental health of health workers. Therefore, I suggest reviewing such papers which been published form similar setting including Nepal. While moving to the methodology section, I felt few improvements can be done. The authors did not talk about peak of epidemics in Gandaki province (period, cases and their comparison with national data). 

We have tried to address the comment in the Introduction section reflecting about the different waves of COVID 19 in Nepal. We have added information from articles which have explored the mental health of health workers during the pandemic. To our knowledge, there are no paper assessing the social impact of COVID 19 in Nepal. In that aspect, we can say that our paper is a novel one. 

We have revised the Methodology section as per your suggestion 

The Gandaki province also followed the same pattern of National COVID-19 infection. Its information is also added briefly in the Introduction section considering the length of the paper. 

The reasons behind choosing 418 sample size been missing. Similarly, for Hosmer and Lemeshow test what is the cut off value you consider the most fit model? How been the interview conducted (Physical or online)?

The methodology section has been revised to address all the suggestions provided on these aspects 

o We have added new section in Methodology section as sample size and sampling technique to clarify this issue regarding sample size.

o The cut-off for goodness of fit and VIF has been added.

o The qualitative interviews were performed physically and its information has been added under Qualitative data collection and analysis 

While reading the paper, I found some typos been occurred. For an example- Consistency form or from Nepali to English.

Thank you for highlighting this issue. We have checked for the typos and corrected them in our document. 

In the result section, consistency in writing the table headings were reported. Therefore, suggested to review similar paper while reporting the table headings and subheadings. Similarly, I suggest you to review the style of reporting of aOR(CI), uOR( CI)- Please refer to other paper.

Thank you for your suggestion. We have revised it. 

You have explored the theme 1, 2, 3 etc. but there is little marriage between quantitative and qualitative information. Therefore, suggested having marriage between qualitative and quantities information. So, I suggest reviewing the similar paper on the mixed method studies.

Thank you for your suggestion. We have revised the Results section linking the findings of both quantitative and qualitative components. 

Authors mentioned about the MoHP expectations were explored but little information were provided on provincial MoHP or federal MoHP. At the point the authors can provide information about how the health system worked or suffered in COVID 19 response in federal context.

The expectations were based on all three levels of government. We have also included this in the manuscript to better clarify the information.

Similar improvements on discussion and conclusion section are required. Please try to incorporate the reviewer’s comments. I further suggest providing all the supplementary materials such as questionnaire, checklist, consent form, reporting guideline etc. These supplementary files can be used as in example. Example: - data was collected through the standard checklist (Supplementary file A).

Thank you. We have made some revisions in the Discussion and Conclusion sections. We have added all supplementary materials in the manuscript as per your suggestion. 

Reviewer 1 Comments and Reply:

While the questionnaire includes many interesting aspects of social influences on depression rate, I miss any question about seeking help for the depression. Especially, when some of the health workers report severe depression, it would be of high interest what kind of treatment they are seeking. Especially for the results and the discussion the treatment and adherence to treatment in Nepal would be highly interesting and should be included. Additionally, depression rates in the general population in Nepal are necessary to compare to the health workers. This could end in an odds ratio or risk ratio calculation and would be an highly informative addition.

Thank you for your valuable feedback. In the online survey (quantitative component) we failed to assess the information about the help seeking behavior of health professionals regarding the treatment of depressive symptoms. During the in-depth interviews (qualitative component), we incorporated the coping mechanism of participants towards the management of social as well as mental impact of COVID-19. In this regards, the participants were more expressive regarding their coping strategy adopted for their social experiences but responded less about the strategies adopted to manage their psychological distress. We have highlighted this concern as a part of our limitation. 

The citation style has to be adapted to PLOS One style.

Thank you. We have revised the citation style as per the Plos One author guideline. 

In Table 6 delete the "0" before Impact over family relation.

Thank you for highlighting this typo. We have removed it. 

Reviewer 2 Comments and Reply:

If you used valid questionnaire, please mention validity and reliability of them , if not please mention it in limitations.

Thank you for this comment. For quantitative survey mostly we used standard validated tool and have added about their validity and reliability measures provided by the past study. We have also added about the pretesting of the tools. In terms of qualitative study, we had prepared the interview guideline based on the finding from the quantitative survey and in consultation with the experts. A brief information about the tool development and piloting has also been added in the Methodology section. Thank you for highlighting this missing aspect of our manuscript. 

Please clarify: What method was used for the qualitative study? What was the way of coding and reaching the themes? What methods are used for the validity of the qualitative study?

Thank you for these comments. We have added some information in the Methodology section under Qualitative data collection and analysis to clarify these issues. 

Abbreviations should be given in full words the first time they are used.

correct some writing errors e.g., font size in the paragraph after table 5 and removing 0 from the word impact in the table 6

Thank you for highlighting these issues. We have corrected them.

---

## [Editor Report · Decision Letter 1]

21 Mar 2023

Social and Mental health impact of COVID-19 Pandemic among Health Professionals of Gandaki Province, Nepal: A mixed method study

PONE-D-22-30030R1

Dear Dr. Chalise,

We’re pleased to inform you that your manuscript has been judged scientifically suitable for publication and will be formally accepted for publication once it meets all outstanding technical requirements.

Kind regards,

Kanchan Thapa, MPH, MPhil

Academic Editor

PLOS ONE

Additional Editor Comments (optional):

Dear Anisha,

Thank you for improvising the papers as per the comments from editors and peer reviewer. Now, I would like to congratulate for making this publication possible which will definitely add value in the field of public health research. The paper will be important assets which represent the provincial aspect in the newly formed federal structure with special focus on public health emergency in Nepal.

Best- Kanchan
---

## [Editor Report · Acceptance letter]

28 Mar 2023

PONE-D-22-30030R1 

Social and Mental health impact of COVID-19 Pandemic among Health Professionals of Gandaki Province, Nepal: A mixed method study 

Dear Dr. Chalise:

I'm pleased to inform you that your manuscript has been deemed suitable for publication in PLOS ONE. Congratulations! Your manuscript is now with our production department. 

Kind regards, 

on behalf of

Mr. Kanchan Thapa 

Academic Editor

PLOS ONE